# Treatment Dilemma in Children with Late-Onset Pompe Disease

**DOI:** 10.3390/genes14020362

**Published:** 2023-01-30

**Authors:** Martha Caterina Faraguna, Viola Crescitelli, Anna Fornari, Silvia Barzaghi, Salvatore Savasta, Thomas Foiadelli, Daniele Veraldi, Matteo Paoletti, Anna Pichiecchio, Serena Gasperini

**Affiliations:** 1Residency in Pediatrics, University of Milano Bicoccca, 20126 Milano, Italy; 2Department of Pediatrics, Fondazione IRCCS San Gerardo dei Tintori, 20900 Monza, Italy; 3Department of Pediatrics and Rare Diseases, Ospedale Microcitemico Antonio Cao, University of Cagliari, 09124 Cagliari, Italy; 4Department of Pediatrics, Fondazione IRCCS Policlinico San Matteo, 27100 Pavia, Italy; 5Neuroradiology Department, IRCCS “C. Mondino Institute of Neurology” Foundation, University of Pavia, 27100 Pavia, Italy

**Keywords:** glycogen storage disease type 2, Pompe disease, late onset, muscle MRI, Enzyme Replacement Therapy

## Abstract

In recent years, there has been a significant increase in the diagnosis of asymptomatic Late-Onset Pompe Disease (LOPD) patients, who are detected via family screening or Newborn Screening (NBS). The dilemma is when to start Enzyme Replacement Therapy (ERT) in patients without any clinical sign of the disease, considering its important benefits in terms of loss of muscle but also its very high cost, risk of side effects, and long-term immunogenicity. Muscle Magnetic Resonance Imaging (MRI) is accessible, radiation-free, and reproducible; therefore, it is an important instrument for the diagnosis and follow-up of patients with LOPD, especially in asymptomatic cases. European guidelines suggest monitoring in asymptomatic LOPD cases with minimal MRI findings, although other guidelines consider starting ERT in apparently asymptomatic cases with initial muscle involvement (e.g., paraspinal muscles). We describe three siblings affected by LOPD who present compound heterozygosis and wide phenotypic variability. The three cases differ in age at presentation, symptoms, urinary tetrasaccharide levels, and MRI findings, confirming the significant phenotypic variability of LOPD and the difficulty in deciding when to start therapy.

## 1. Introduction

Glycogen storage disease type II, or Pompe disease, is a rare autosomal inborn error of the metabolism, due to a deficiency of acid α-glucosidase, an enzyme that breaks down glycogen.

It has been historically classified on the basis of age at presentation as Infantile-Onset Pompe Disease (IOPD) and Late-Onset Pompe Disease (LOPD) [1]. While IOPD is more homogenous in age, symptoms, and signs at presentation with cardiomyopathy (classic IOPD), LOPD presents a wide range of genotypes and phenotypes. In fact, it varies on the basis of the mutations, but, within families with the same mutations, significant clinical differences have been described [2,3,4].

In the last decade, an increasing number of pre-symptomatic LOPD cases has been identified thanks to screening within the family of a diagnosed individual, improved awareness of the disease, and the introduction of expanded Newborn Screening (NBS).

The recognition of pre-symptomatic cases has raised the dilemma of when to start treatment. Therapy for Pompe disease consists of intravenous replacement therapy (ERT) with a human recombinant enzyme, alfa-glucosidase [5,6]. ERT improves walking distance, quality of life [7], pulmonary function stabilization in LOPD cases [5], increases survival, reduces ventilation in IOPD, and improves cardiomyopathy [6]. ERT presents side effects, including local (pain, edema, and erythema) and systemic reactions (anaphylaxis) [7] Furthermore, long-term therapy causes seroconversion, with 95% of cases becoming IgG positive. This is frequent in IOPD patients and especially in CRIM (cross-reactive immunologic material status) negative ones with very low residual enzyme activity. Exceptionally high titers of persistent antibodies may occur, and this can activate the complement cascade, neutralizing the recombinant enzyme. In these rare circumstances, there is a decline in the therapeutic response to ERT, and infusion reactions may be observed [8,9,10,11]. 

New drugs are being developed, such as avalglucosidase alfa [12], cipaglucosidase, and miglustat, as well as gene therapy trials that are ongoing [13,14]. 

We describe three siblings affected by Late-Onset Pompe Disease (LOPD) who present a compound heterozygosis in two variants, one of which has not been previously described in the literature, and who differ in age at presentation, symptoms, urinary tetrasaccharide levels, and muscle MRI findings, confirming the significant phenotypic variability of LOPD and the difficulty in choosing when to start therapy. Patient characteristics are reported in Table 1 and Figure 1. 

## 2. Case Report

A 7-month-old infant (patient 1) with a recent diagnosis of Pompe disease (infantile non-classic form) was brought to our attention. He was born at term after an uncomplicated pregnancy. Hypotonia of the head was noted at discharge from the hospital on the third day of life. At three months of age, his parents brought him to the emergency department of a peripheral hospital, where a marked hypotonia of the head and trunk was noted. He was sent home with a neurological follow-up visit, which confirmed hypotonia of the flexor muscles of the neck, plagiocephaly, and suture prominence with normal auxological parameters. 

The infant’s parents are of non-consanguineous Egyptian origin. Family history revealed muscle fatigability and cramps in the mother from a young age; creatine phosphokinase (CPK, 134 U/L, n.v. 24–190) and electromyography were normal. He has a healthy older sister (patient 2) and brother (patient 3). He never suffered from respiratory infections.

Blood samples resulted normal, except for a slight raise in CPK (275 mU/mL at five months of age; 316 mU/mL at six months; normal value, 24–190 U/L).

The boy underwent a 3-D head computerized tomography to exclude craniosynostosis, which was normal; and a CGH array for Prader-Willi and congenital myasthenia, which were also normal; a dry blood spot test for Pompe disease, which resulted in borderline values (1.9 µmol/L/h; normal value > 2); and NGS sequencing of the GAA gene (OMIM 232300) was performed on peripheral blood. Written consent was acquired from the parents. A combined heterozygosis was identified: GAA (NM_000152.3):c.2284G>A p.(Glu762Lys) and GAA (NM_000152.3):c.1994G>A p.(Gly665Glu).

The c.2284G>A variant has already been described [15,16] and is classified as a Variant of Unknown Significance (VUS); the same variant was found in the father, who presented normal enzyme activity (3.8 micromol/L/h; normal value > 2.0). The c.1994G>A variant, which was identified in the mother, who presented normal enzyme activity (2.2 micromol/L/h), has not yet been described in the literature; according to the American College of Medical Genetics (ACMG), it is classified as probably pathogenetic. Such a hypothesis has been confirmed by in silico predictions (Mutation Taster, SIFT, and Polyphen-2). An electrocardiogram and echocardiography were normal. Brain MRI showed a slight increase in the subarachnoid spaces in the anterior temporal, insular, and frontoparietal regions bilaterally and a thin corpus callosum. 

During his first visit to our center, the 7-month-old boy made eye contact, smiled, and was able to sit up by himself. He turned in a clockwise fashion and presented only a slight hypotonia of the upper back. No other signs of Pompe disease were present. Blood tests were normal except for a slight raise in CPK (372 U/L, n.v. < 190), lactate dehydrogenase (LDH, 342 U/L, n.v. 120–300), aspartate aminotransferase (AST, 40 U/L, n.v. < 40), and alanine transaminase (ALT, 24 U/L, n.v. < 41). Creatine Kinase Muscle Band (CK-MB, 11.8 ng/mL, n.v. < 6.22), and pro-B-type natriuretic peptide (pro-BNP, 31 pg/mL, n.v. < 320) were not altered. The measurement of acid maltase level on leukocytes was slightly decreased (0.33, n.v. > 0.35). 

Considering the unknown significance of the c.2284G>A variant, the borderline value of acid maltase on leukocytes and enzyme activity on dry blood spot, and the normality of cardiac assessment, a urinary tetrasaccharide was dosed to exclude a pseudo-deficiency, which resulted in an increase (3.21 mmol/mol/cr, compared with the normal value of 0.08–1.37).

Meanwhile, the patient’s brother and sister underwent extensive testing. No symptoms regarding the 8-year-old sister were reported (patient 2). She was admitted to the pediatric department at 7 years old because of a urinary tract infection, and during an abdomen ultrasound, mild steatosis was reported. As a screening for Pompe disease, she underwent dosing of enzyme activity on dry blood, which resulted in low activity (1.2 µmol/L/h, n.v. > 2), and sequencing of the GAA gene, which found the same mutations as patient 1. A neurologic visit, electromyography, and echocardiography were performed, which were all normal. Lower limb muscles MRI demonstrated diffuse, slight fat replacement mainly at the levels of the medial and posterior compartments of the thigh, as well as in the gluteus maximus; no edematous changes were shown on the Short TI Inversion Recovery Sequence (STIR). Further slight fat replacement was evident in the triceps surae with correspondent “edema-like” STIR positive abnormalities (Figure 1).

She was therefore referred at our center. She reported no symptoms except for a slight fatigability in the last four months. She played sports twice a week and did not present muscle cramps or dyspnea. She did not present any sign of muscle weakness except for a difficulty in whistling and blowing properly. Urinary tetrasaccharide (Glc4) was in the normal range (0.78 mmol/mol Cr, n.v. 0.08–1.37), CPK 153 U/L (n.v. < 170), AST 25 U/L (n.v. < 32), LDH 357 U/L (n.v. 120–200), CK-MB 3.41ng/mL (n.v < 4.88), and NT-proBNP 79 pg/mL (n.v. < 145). The 6-Minute Walking Test (6MWT) and spirometry results were normal.

The older brother (patient 3) underwent screening for Pompe disease when he was 10 years old. His history was negative. Echocardiography and neurologic assessment results were normal; enzyme activity on the dry blood spot result was borderline (1.2 µmol/L/h, n.v. > 2); and GAA sequencing revealed the same mutations as the younger siblings. An abdomen ultrasound, electromyography, and muscle MRI were all normal (Figure 2), with no signs of selective fat replacement or edema on STIR. He was also referred to our center. He played sports three times a week and reported no symptoms. He did not present any signs of myopathy. Blood tests (CPK 183U/L, AST 20U/L, LDH 252U/L, pro-BNP 52pg/mL, and CK-MB 3.3ng/mL), urinary tetrasaccharide (0.69 mmol/mol Cr, n.v. 0.08–1.37), 6MWT, and spirometry were all normal.

All three siblings present the same mutations, one classified as probably pathogenic and one as of unknown significance. Patient 1 was symptomatic only during the first three months of age (slight hypotonia of the upper back) and presents an increase in urinary Glc4 and CPK. Patient 2 refers to fatigability and has muscle MRI abnormalities consistent with LOPD (although not specific). Patient 3 is asymptomatic and presents a normal muscle MRI and Glc4. Echocardiography is normal in all cases. 

At eight years and six months, patient 2 started Enzyme Replacement Therapy (ERT) with human recombinant acid, a-glucosidase 20 mg/kg/dose, which was administered every other week. Patient 3 is evaluated every 4–6 months and undergoes muscle MRI every 12 months, while patient 1 is seen every 3–4 months and will shortly start ERT as well.

## 3. Discussion

The three siblings reported differ in age at diagnosis, symptoms, and findings; patient 2, is the only symptomatic one and presents signs of initial disease involvement at muscle MRI, whereas patient 1 was initially symptomatic and presents an increase in CPK and Glc4. The dilemma is determining when to start Enzyme Replacement Therapy (ERT) in each patient. 

European countries’ guidelines are all 10 years old and do not address the problem of asymptomatic patients [17,18,19]. There are no Italian guidelines to date.

With the advent of newborn screening programs that include some lysosomal storage diseases such as Pompe disease, the clinical dilemma about when to start ERT in LOPD has significantly expanded. While there is agreement on starting ERT early in IOPD to change the natural evolution dramatically, even in CRIM-negative patients (mean age of 12 days in Taiwan [20]), there is no consensus on a very early diagnosis of cases of LOPD, which may have been diagnosed at 60 years old after 10 years of symptoms. The decision to start ERT to stop the progression of a disease that might be asymptomatic for decades is hard to take, especially in patients apparently without clinical signs, due to very high costs (economic and social), the development of antibodies, and life-long intravenous infusions [21,22].

The ethical problem is detecting the disease and not being able to prevent the cellular and subcellular damages before they occur as organ damage. In fact, it has been recently acknowledged that Pompe disease is a multisystem disorder (not only muscular) and that other intracellular mechanisms of autophagy are involved [23,24,25].

In 2017, the Pompe Disease Newborn Screening Working Group suggested that the best morphologic results from ERT may be achieved when treatment is started while patients have measurable signs of disease but are still clinically asymptomatic, as preventive medicine. Additional studies are needed to support or refute these findings [26]. 

The European Consensus of 2017 for starting and stopping ERT underlines that there is currently insufficient evidence to support starting ERT in pre-symptomatic patients [27]. It is recommended that these patients are monitored every 6–12 months to identify disease progression early; treatment should not be started in the absence of both skeletal muscle weakness and respiratory involvement. This recommendation to monitor pre-symptomatic patients includes patients who experience fatigue or myalgia, have elevated creatine kinase levels, or show minimal pathological findings on magnetic resonance imaging or muscle biopsy in the absence of skeletal muscle weakness and/or respiratory involvement. The consensus group hypothesizes that asymptomatic patients may already be losing muscle mass, which they may not be able to regain. Thus, it is important to obtain more evidence to assess whether such patients would benefit from treatment, but the high costs of drugs may hamper such studies.

Brazilian guidelines suggest considering starting treatment in asymptomatic patients with a typical vacuolar muscle biopsy; imaging findings are not considered [10].

The Middle East and North Africa group considers ERT unnecessary in pre-symptomatic patients with no signs or symptoms, although these patients should be monitored every 6 months and ERT initiated if there is evidence of clinical deterioration in muscle or pulmonary function. Pre-symptomatic patients who have abnormal muscle imaging or muscle biopsy and patients without clinical signs but with MRI abnormalities in muscles not traditionally tested (e.g., paraspinal muscles) should be considered for treatment on a case-by-case basis [28].

Conventional muscle MRI sequences (T1-weighted and STIR) identify fat replacement and increased water content and are applied to detect muscle involvement in Pompe disease, as well as in other neuromuscular diseases. The amount of fat replacement correlates well with muscle function testing and can be considered a good outcome measure in patients with LOPD [29]. Muscle MRI can therefore be used to monitor evolution in treated and nontreated LOPD patients [30], in a context in which fat replacement progresses over time in untreated subjects [31,32]. 

Early and subtle signs of muscle fat replacement (especially in the paraspinal and abdominal muscles) without any clinical symptom of muscle weakness or abnormality at clinical examination, functional and respiratory tests, have been observed in LOPD cases, suggesting that the process of muscle degeneration had already started in asymptomatic subjects. These findings could potentially be considered before deciding to start ERT in a patient with clinically asymptomatic LOPD. It is therefore essential to repeat muscle MRI evaluations in such cases to detect early signs of the disease [33].

## 4. Conclusions

We describe three siblings affected by LOPD who present a compound heterozygosis with a variant that has not yet been described in the literature and might be classified as probably pathogenetic. The three cases differ in age at presentation, symptoms, urinary tetrasaccharide levels (Glc4), and MRI findings, confirming the significant phenotypic variability of LOPD in the same family.

In recent years, there has been a significant increase in the diagnosis of asymptomatic LOPD patients, who are detected via family screening or NBS. The dilemma is when to start ERT, considering its important benefits in terms of muscle but also its very high cost, risk of side effects, and long-term immunogenicity. 

Muscle MRI is accessible, radiation-free, and fairly reproducible in evaluating disease involvement (i.e., fat replacement and edema-like changes); therefore, it can be considered an important instrument for supporting clinical diagnosis and evaluating longitudinally patients with LOPD, especially in asymptomatic cases. European guidelines suggest monitoring asymptomatic LOPD cases with minimal MRI findings, although other guidelines consider starting ERT in asymptomatic cases with initial muscle involvement (e.g., paraspinal muscles).

Muscle MRI can be considered a potential useful biomarker not only for diagnosis and follow-up but also to define when to start therapy. 

## Figures and Tables

**Figure 1 genes-14-00362-f001:**
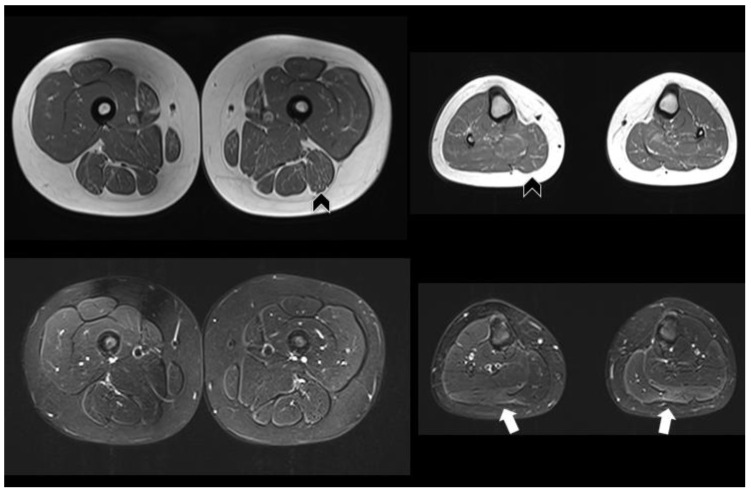
Muscle MRI of the 8-year-old sister (patient 2). Axial T1-weighted images (**top**) show slight fat replacement mainly at the level of the posterior and medial compartments of the thigh and of the triceps surae at the level of the leg, bilaterally (black arrowheads). STIR images (**bottom**) only show mild edema-like changes at the level of the gastrocnemius lateralis, bilaterally (arrows).

**Figure 2 genes-14-00362-f002:**
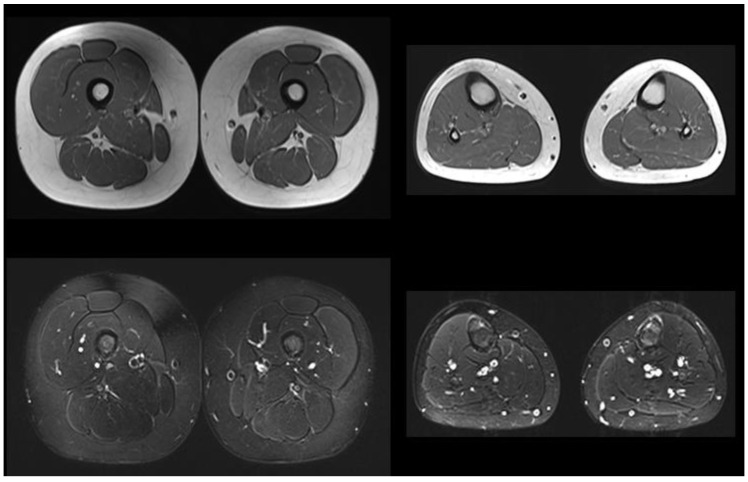
Lower limb muscles MRI of the older brother when 10 years old; T1-weighted (**top**) and STIR images (**bottom**) are shown. No significant fat replacement or edema changes are detected at the thigh and leg levels, bilaterally.

**Table 1 genes-14-00362-t001:** Comparison of clinical symptoms and signs, blood results, and imaging results between the three siblings at the first evaluation at our center. Available data on the mother is also reported.

	Patient 1 7 Months Old	Patient 2 8 Years Old	Patient 3 10 Years Old	Mother
Symptoms	None	Slight fatigability in the previous four months	None	Muscle fatigability and cramps from a young age
Signs of disease are present at evaluation	Slight hypotonia of the upper back	Difficulty in whistling and blowing properly	None	None
Serum Creatine Phosphokinase (CPK, U/L) Normal value < 190	372	153	183	134
Lactate Dehydrogenase (LDH, U/L) Normal value 120–300	342	357	242	Not available
Aspartate Transaminase (AST, U/L) Normal value < 40	40	25	20	Not available
Alanine Transaminase (ALT, U/L) Normal value < 41	24	17	25	Not available
Creatine Kinase Muscle Band (CK-MB, ng/mL) Normal value < 6.22	11.8	3.41	3.3	Not available
Pro-B-type Natriuretic Peptide (pro-BNP, pg/mL) Normal value < 320	31	79	52	Not available
Acid maltase activity on leukocytes Normal value > 0.35	0.33	Not performed	Not performed	Not performed
Enzyme activity on a dry blood spot (µmol/L/h) Normal value > 2	1.9	1.2	1.2	
Urinary Tetrasaccharid (Glc4, mmol/mol/cr) Normal value 0.08–1.37	3.21	0.78	0.69	Not performed
GAA sequencing	c.2284G>A p.(Glu762Lys); c.1994G>A p.(Gly665Glu)	c.2284G>A p.(Glu762Lys); c.1994G>A p.(Gly665Glu)	c.2284G>A p.(Glu762Lys); c.1994G>A p.(Gly665Glu)	c.1994G>A p.(Gly665Glu)
Electromyography	Normal	Normal	Normal	Normal
Heart ultrasound	Normal	Normal	Normal	Not performed
Spirometry	Not performed	Normal	Normal	Not performed
Six Minute Walking Test	Not performed	Normal	Normal	Not performed
Muscle MRI	Not performed	Pathologic (Figure 1)	Normal	Not performed

## Data Availability

Data is available upon request to the corresponding author.

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
