# Peer review of "Treatment Dilemma in Children with Late-Onset Pompe Disease"

_genes, 2023, doi:10.3390/genes14020362_

Round 1

Reviewer 1 Report

The manuscript presents a case of 3 siblings with the same genetic GAA variants, but presenting with different symptoms: one of them was symptomatic during the first three months of life, the second with a LOPD and the third was asymptomatic. The therapy start issue in the second part is extensively discussed in terms of disease heterogenicity and therapy start point. Further, the application of MRI in asymptomatic patients is highlighted.

The introduction provides sufficient background and dilemmas regarding when to start treatment, side effects and new therapies are clearly presented and understandable. The case report is detailed and laboratory value, as well as diagnostic parameters, are provided. In the discussion, the problem is well elucidated and the guidelines in different countries are reviewed. MRI images indicate that muscle degeneration starts already early. Conclusions are clear and support results and discussion.

Commentts:

Figures seem to be cut. 

Figure 1, top images-I would mark changes with arrows.

Author Response

Thank you very much for your positive comments. We have modified Figure 1 and placed it correctly in the text.

Reviewer 2 Report

I have reviewed an original article titled: “Treatment dilemma in children with Late Onset Pompe Disease”. There are some comments and concerns that the authors might want to further discuss and follow-up. These concerns and comments are listed as follow:

1. a. Maybe I have overlooked, after reading the whole manuscript, I still cannot find what exactly are the 2 compound heterozygous variants of subjects 1,2 and 3 have. It was wonder can the authors provide the details of these variants found from these subjects. Or is that because no ethical approval for publication was obtained from the family? Or other specific reasons?

1. b. as a follow-up of a, please kindly state the variant in a format recommended by the ACMG guideline at least the first time using it in the manuscript. E.g. BRAF(NM_004333.6):c.1799T>G (p.Val600Gly).

1. c. It seems that classifying the pathogenicity of a variant according to ACMG guideline is better than that to the predictive bioinformatics software (which usually used as supporting evidence). What is the pathogenicity according to the ACMG guidelines? Which ACMG evidence this variant fitted-in? e.g. PS1, PVS1 or PM1 etc.? 

1. d. What in silico predictions software the authors have used in this study (line 84)? Can the authors name them?

2. To my understanding, the authors is trying to present the scenario as having ERT too late would lead to some damages at cellular or subcellular level. While on the other hand, administration of ERT to the patients also have some side-effects, medical consequence, or implications on the patients. Thus, early administration is also not a very good choice to (at least some) patients.

In reality, despite not starting ERT, most (if not all) organizations around the world suggested to keep a close monitor on the asymptomatic patients at a regular short interval of biennial or annually, and ERT will be administered once detectable deterioration was noticed. Therefore, from these points of view, will these suggestions by these organization be considered as already “strike a balance” between the cost (in terms of social, economic and medical) and the damaging effect of the disease as mentioned by the authors? In this sense, I do not see any “dilemma” here or I think the “dilemma” as mentioned by the authors is not that much. Or can the authors elaborate further on that?

3. Please kindly explain what genetic test has been done on these subjects. Whole exome sequencing? Sanger sequencing? Or what kind of test these subjects have been done?

4. As the proband in this study, is it possible to have any medical or clinical images or data showing the severity or manifestation observed in proband, especially hypotonia was observed. Furthermore, it was also mentioned that “Electrocardiogram and echocardiography resulted normal. Brain MRI showed a slight increase of the subarachnoid spaces in the anterior temporal, insular and frontoparietal region bilaterally and a thin corpus callosum.” (Line 85-87) These are also important results data to be presented. Please kindly provide these data for subject 1. Moreover, it was also mentioned a few experiments has been done, such as aCGH, genetic test (despite it is not known what test it actually is). Will there be any chance these results data and/or figures be presented asl well?

5. Maybe I have overlooked again, please provide a Sanger sequencing of the variants observed from the subjects 1,2, 3 and their biological father and mother. Thank you.

6. Can the authors please confirm or further explain the following sentences?

6. a. The first variant has already been described and is classified as variant of unknown origin, inherited from the father, who presented normal enzyme activity (3.8 micromol/L/h, normal value > 2.0). (line 79-82)

I guess the authors meaning that variant was inherited from father, while that variant was a de novo mutation found in father (i.e. the genetic of the grandparents were also tested and cannot found this variant.)  

6. b. What does “last months” means in line 121? Maybe using “X months after her visit” would be a better approach if the authors would like to refer to a particular month?

6. c. “European countries guidelines are all 10 years old and do not address the problem of asymptomatic patients.” (Line 158). So it means the European countries will treat paediatric patients younger than 10 years-old or older than 10 years-old?

7. As each of the parents possess one mutation and given their physical condition as described in the manuscript, are they considered as the patients of Pompe disease as well? If yes, Why? If not, why not?

8. Without properly knowing which variant Line 96 is referring to, it is very much hard to have any knowledge exchange for that paragraph.

9. Maybe I have overlooked, it seems that there were no any statement on the ethical approval for these studies.

10. a. It was mentioned that “All three siblings present the same mutations, one classified as pathogenic, one as of unknown origin.” (Line 143-144). To my understanding, “pathogenic” and “unknown origin” are not like terms for comparison. To my poor understanding of genetic pathogenicity, “pathogenic”, “likely pathogenic”, “benign”, “Variant of uncertain significance (VUS)” are the proper terms to be used. Its originality is not related to its pathogenicity.

10. b. Furthermore, maybe I misunderstood, in the main text it seems that it was mentioned to be “it may be classified as probably pathogenetic” (Line 84) while in discussion it was “upgraded” to pathogenic (line 143-144). Maybe can the authors fix that or using ACMG guideline as the pathogenicity classification criteria?

11. Also, any statements on obtaining written consent from the subjects or their guardians to participate in this study?

12. I know the authors are clinician and experts in the field, however, not all readers are as knowledgeable about Pompe disease, thus, many short forms like LDH, AST, ALT, CK-MB, Pro-BNP etc. might not be commonly known to some readers or might have them guessed wrong. Thus, please provide the full name of these short form at the first time using them or list all of them out somewhere in the manuscript.

13. There are quite a number of clinical and lab results from mother, subject 1, 2 and 3.  It was wondered if it is possible to tabulate the lab test results for easier reading and comparison?

14. Please kindly provide reference for the following sentences:

14. a. ERT presents side effects, including local (pain, edema, erythema) 46 and systemic reactions (anaphylaxis). (line 46)

14.b. Further so, long term therapy causes seroconversion, 95% of cases becoming IgG positive. (line 48)

14.c. In 2017, the Pompe Disease Newborn Screening Working Group suggested that the best morphologic results from ERT may be achieved when treatment is started while patients have measurable signs of disease but are still clinically asymptomatic as preventive medicine. (line 174-176)

14.d. Early and subtle signs of muscle fat replacement (especially in the paraspinal and abdominal muscles) without any clinical symptom of muscle weakness or abnormality at clinical examination, functional and respiratory tests, have been observed in LOPD cases, (Line 206-208).

Author Response

I have reviewed an original article titled: “Treatment dilemma in children with Late Onset Pompe Disease”. There are some comments and concerns that the authors might want to further discuss and follow-up. These concerns and comments are listed as follow.

  1. Maybe I have overlooked, after reading the whole manuscript, I still cannot find what exactly are the 2 compound heterozygous variants of subjects 1,2 and 3 have. It was wonder can the authors provide the details of these variants found from these subjects. Or is that because no ethical approval for publication was obtained from the family? Or other specific reasons)

Thank you for you comment. We have added the variants to the text. Approval from the family was obtained.

  1. b. as a follow-up of a, please kindly state the variant in a format recommended by the ACMG guideline at least the first time using it in the manuscript. E.g. BRAF(NM_004333.6):c.1799T>G (p.Val600Gly).

We have added the variants according to the ACMG guidelines.

  1. c. It seems that classifying the pathogenicity of a variant according to ACMG guideline is better than that to the predictive bioinformatics software (which usually used as supporting evidence). What is the pathogenicity according to the ACMG guidelines? Which ACMG evidence this variant fitted-in? e.g. PS1, PVS1 or PM1 etc.?

According to ACMG classification, the c.2284G>A variant is classified as Variant of Unknown Significance (VUS). The c.1994G>A variant is classified as Probably Pathogenetic.

  1. d. What in silico predictions software the authors have used in this study (line 84)? Can the authors name them?

The c.1994G>A variant is classified as Probably Pathogenetic according to ACMG. Such hypothesis has been confirmed by in silico predictions (Mutation Taster, SIFT, Polyphen-2).

  1. To my understanding, the authors is trying to present the scenario as having ERT too late would lead to some damages at cellular or subcellular level. While on the other hand, administration of ERT to the patients also have some side-effects, medical consequence, or implications on the patients. Thus, early administration is also not a very good choice to (at least some) patients.

In reality, despite not starting ERT, most (if not all) organizations around the world suggested to keep a close monitor on the asymptomatic patients at a regular short interval of biennial or annually, and ERT will be administered once detectable deterioration was noticed. Therefore, from these points of view, will these suggestions by these organization be considered as already “strike a balance” between the cost (in terms of social, economic and medical) and the damaging effect of the disease as mentioned by the authors? In this sense, I do not see any “dilemma” here or I think the “dilemma” as mentioned by the authors is not that much. Or can the authors elaborate further on that?

Thank you for your comment. The decision on when to start ERT is not straightforward, as clinicians must consider symptoms, laboratory results, mutations and imaging findings. Furtherly, symptoms are often difficult to quantify, especially in young children. As reported in this paper, three siblings within a same family present different scenarios of the disease. The term “dilemma” suggests that the decision is not easy to take and many factors must be considered.  Finally, the decision of starting ERT must be discussed and shared with the family.

  1. Please kindly explain what genetic test has been done on these subjects. Whole exome sequencing? Sanger sequencing? Or what kind of test these subjects have been done?

We are sorry to not have specified it initially. We have modified the text. NGS analysis of the GAA gene (OMIM 232300) was performed.

  1. As the proband in this study, is it possible to have any medical or clinical images or data showing the severity or manifestation observed in proband, especially hypotonia was observed. Furthermore, it was also mentioned that “Electrocardiogram and echocardiography resulted normal. Brain MRI showed a slight increase of the subarachnoid spaces in the anterior temporal, insular and frontoparietal region bilaterally and a thin corpus callosum.” (Line 85-87) These are also important results data to be presented. Please kindly provide these data for subject 1. Moreover, it was also mentioned a few experiments has been done, such as aCGH, genetic test (despite it is not known what test it actually is). Will there be any chance these results data and/or figures be presented asl well?

No clinical images are available. Unfortunately, neither are the images of the brain MRI. The part concerning the genetic testing has been modified (see points 1-3).

  1. Maybe I have overlooked again, please provide a Sanger sequencing of the variants observed from the subjects 1,2, 3 and their biological father and mother. Thank you.

The part concerning the genetic testing has been modified (see points 1-3)

  1. Can the authors please confirm or further explain the following sentences?
  2. a. The first variant has already been described and is classified as variant of unknown origin, inherited from the father, who presented normal enzyme activity (3.8 micromol/L/h, normal value > 2.0). (line 79-82)

I guess the authors meaning that variant was inherited from father, while that variant was a de novo mutation found in father (i.e. the genetic of the grandparents were also tested and cannot found this variant.) 

Line 79: the variant was inherited from the father. We do not know whether it is a de novo mutation in the father, no further relatives were tested (e.g. the boy’s grandparents were not tested).

  1. b. What does “last months” means in line 121? Maybe using “X months after her visit” would be a better approach if the authors would like to refer to a particular month?

We have modified the text.

  1. c. “European countries guidelines are all 10 years old and do not address the problem of asymptomatic patients.” (Line 158). So it means the European countries will treat paediatric patients younger than 10 years-old or older than 10 years-old?

European guidelines are not recent (more than ten years old); children of any age are considered for treatment.

  1. As each of the parents possess one mutation and given their physical condition as described in the manuscript, are they considered as the patients of Pompe disease as well? If yes, Why? If not, why not?

The parents present a single mutation for Pompe disease, which is an autosomal recessive disorder. Therefore they are not considered patients, they are heterozygous carriers.

  1. Without properly knowing which variant Line 96 is referring to, it is very much hard to have any knowledge exchange for that paragraph.

We are sorry for not being clear. We have modified the text.

  1. Maybe I have overlooked, it seems that there were no any statement on the ethical approval for these studies.

We have added a statement.

  1. a. It was mentioned that “All three siblings present the same mutations, one classified as pathogenic, one as of unknown origin.” (Line 143-144). To my understanding, “pathogenic” and “unknown origin” are not like terms for comparison. To my poor understanding of genetic pathogenicity, “pathogenic”, “likely pathogenic”, “benign”, “Variant of uncertain significance (VUS)” are the proper terms to be used. Its originality is not related to its pathogenicity.

We are sorry for not being precise. We have modified the text.

  1. b. Furthermore, maybe I misunderstood, in the main text it seems that it was mentioned to be “it may be classified as probably pathogenetic” (Line 84) while in discussion it was “upgraded” to pathogenic (line 143-144). Maybe can the authors fix that or using ACMG guideline as the pathogenicity classification criteria?

We are sorry for the mistake. We have modified the text.

  1. Also, any statements on obtaining written consent from the subjects or their guardians to participate in this study?

A statement has been added.

  1. I know the authors are clinician and experts in the field, however, not all readers are as knowledgeable about Pompe disease, thus, many short forms like LDH, AST, ALT, CK-MB, Pro-BNP etc. might not be commonly known to some readers or might have them guessed wrong. Thus, please provide the full name of these short form at the first time using them or list all of them out somewhere in the manuscript.

Thank you for your comment. We have modified the text.

  1. There are quite a number of clinical and lab results from mother, subject 1, 2 and 3. It was wondered if it is possible to tabulate the lab test results for easier reading and comparison?

Thank you for your suggestion. We have created Table 1.

  1. Please kindly provide reference for the following sentences:
  2. a. ERT presents side effects, including local (pain, edema, erythema) 46 and systemic reactions (anaphylaxis). (line 46)

Reference has been added [7].

14.b. Further so, long term therapy causes seroconversion, 95% of cases becoming IgG positive. (line 48)

Reference is provided at the end of the paragraph [8-11]

14.c. In 2017, the Pompe Disease Newborn Screening Working Group suggested that the best morphologic results from ERT may be achieved when treatment is started while patients have measurable signs of disease but are still clinically asymptomatic as preventive medicine. (line 174-176)

Reference is present at the end of the paragraph [26]

14.d. Early and subtle signs of muscle fat replacement (especially in the paraspinal and abdominal muscles) without any clinical symptom of muscle weakness or abnormality at clinical examination, functional and respiratory tests, have been observed in LOPD cases, (Line 206-208).

Reference is present at the end of the paragraph [33]
